# What do people think of intuitive eating? A qualitative exploration with rural Australians

Nina Van Dyke[1]*, Michael Murphy[2], Eric J. Drinkwater[3¤]

1 Mitchell Institute, Victoria University, Melbourne, Victoria, Australia, 2 MM Research, Melbourne, Victoria, Australia, 3 School of Exercise and Sport Science, Charles Sturt University, Bathurst, New South Wales, Australia

¤ Current address: Centre for Sport Research, School of Exercise & Nutrition Sciences, Deakin University, Geelong, Victoria, Australia

* nina.vandyke@vu.edu.au

**Data Availability Statement:** De-identified transcripts will be considered upon request. Due to the nature of the data (i.e. a small number of focus group participants from a single geographic area), it is very difficult to anonymize the data. In addition,

## Abstract

Evidence supports that intuitive eating is associated with many indicators of positive physical and mental health, with more recent longitudinal studies establishing causality. Most research, however, comprises either survey data or clinical trials. This study attempts to fill this evidentiary gap by using a qualitative methodology to explore people's understandings and reactions to intuitive eating, including perceived barriers and enablers to implementation. Three focus group discussions were conducted in a non-metropolitan region of Victoria, Australia, with a total of 23 participants. Focus group transcripts were thematically analysed using an inductive descriptive approach within a constructionist perspective. Findings indicate that the concept of intuitive eating was either unknown or misunderstood. Once intuitive eating was explained, most responses to implementing intuitive eating were negative. Participants felt that having complete choice around what they ate was unlikely to equate to a healthy or balanced diet, at least in the short term. They also argued that because everyday life was not intuitive in its structures, it would be difficult to eat intuitively. Despite these difficulties, participants appreciated that if they were able to overcome the various barriers and achieve a state of intuitive eating, they anticipated a range of long-term benefits to health and weight management. For intuitive eating to become a viable public health approach, this research suggests that intuitive eating needs to be much more widely publicised and better explained, and perhaps renamed. More significantly, people would need assistance with how to eat intuitively given the barriers identified.

## Introduction

The evidence is now quite robust that intuitive eating is associated with many indicators of positive physical and mental health [1, 2], with more recent longitudinal studies beginning to establish causality [3]. Intuitive eating is an approach to eating that focuses on responding to innate hunger and satiety signals.

The concept was originally developed by two nutritional counsellors in 1995 in response to concerns about clients who were distressed by their bodies and inability to lose weight

the participants did not provide explicit consent for the transcripts to be shared publicly.

**Funding:** NV and ED received funding from Charles Sturt University - Faculty of Education Research Development Fund for this study. URL: https://www.csu.edu.au/. In addition, the Social Research Centre conducted the focus groups at cost. URL: https://srcentre.com.au/. The funders had no role in study design, data collection and analysis, decision to publish, or preparation of the manuscript.

**Competing interests:** The authors have declared that no competing interests exist.

through traditional nutritional manipulations [4]. Tribole and Resch (1995) developed 10 principles of intuitive eating: (1) reject the diet mentality; (2) honor your hunger; (3) make peace with food; (4) challenge the food police; (5) discover the satisfaction factor; (6) feel your fullness; (7) cope with your emotions with kindness; (8) respect your body; (9) movement–feel the difference; and (10) honor your health–gentle nutrition. In 2006, Tylka developed the Intuitive Eating Scale (IES), which measures individuals' tendency to follow their physical hunger and satiety cues when determining when, what, and how much to eat. It was based on the work of Tribole and Resch (1995) and consisted of three sub-scales: unconditional permission to eat, eating for physical rather than emotional reasons, and reliance on internal hunger and satiety cues [5]. In 2013, this scale was refined to include an additional component: body-food congruence [6].

Most intuitive eating research consists of either survey data testing correlations between intuitive eating and various health indicators, or clinical trials examining the impact of intuitive eating interventions on changes in health measures, mostly amongst women with Body Mass Index (BMI) >25kg/m$^2$ [1, 2, 7]. This research indicates that those who practice intuitive eating tend to have better physical and mental health, and individuals with BMI >25kg/m$^2$ who undergo an intuitive eating intervention have positive outcomes after completing the program. However, we know much less about whether the general population has ever heard of intuitive eating or its principles, what they think of such an approach, or whether they might consider adopting such an approach [8]. If most people believe that intuitive eating is a bad idea or too difficult to implement, it may have limited value as a public health approach to improving health. This study is an attempt to fill this gap in the evidence by using a qualitative methodology to explore people's understandings and reactions to intuitive eating, including perceived barriers and enablers to implementation.

## Background

### Intuitive eating and physical and mental health

Academic research on intuitive eating is now 25 years old, with the first paper published in 1998 [9]. Early studies tended to focus on the potential role of intuitive eating in combating the "obesity crisis". Soon, however, it became clear that, although higher scores on intuitive eating measures were indeed associated with lower BMI, even stronger associations existed between intuitive eating and an array of psychological and psycho-social measures, such as psychological distress, depression, body image, and disordered eating. Indeed, most clinical studies testing an intuitive eating intervention (with, generally, a group of women with BMI >25kg/m$^2$ found small and sometimes short-lived reductions in weight or BMI, but stronger improvements in participants' feelings about themselves and their bodies [1, 7].

As the focus for improving public health has moved away from reducing obesity per se towards improving health and wellbeing, so too has the intuitive eating literature focus shifted. The evidence is now firmly established that eating intuitively is associated with an array of positive physical and mental health indicators [2]. Moreover, recent longitudinal research found that the causal direction is from intuitive eating to health outcomes. A study of adolescents in the Minneapolis/St Paul area in the U.S. found that greater baseline levels of intuitive eating and increases in intuitive eating from baseline to follow-up were both associated with lower odds of high depressive symptoms, low self-esteem, high body dissatisfaction, unhealthy weight control behaviors (e.g., fasting; skipping meals), extreme weight control behaviors (e.g., taking diet pills; vomiting), and binge eating at eight-year follow-up [3]. These findings make a clear argument for encouraging the adoption of intuitive eating to improve population health and wellbeing.

## Low rates of intuitive eating

Rates of intuitive eating, however, appear to be low. Estimates vary, depending on the population and the operationalisation of intuitive eating. The only general population statistics we found–a survey of rural Australians—found the rates of intuitive eating to be 9% for women and 26% for men, using a cut-off of 4.0 or higher (out of 5) on the Tylka Intuitive Eating Scale—a cut-off suggested by Tylka [10]. A study of 419 women who had given birth within the previous 6–48 months found that 32% were classified as intuitive eaters, with intuitive eating defined as a score above 3.5 on the same Tylka Intuitive Eating Scale [11]. Much higher rates of intuitive eating were found in a survey of young adults. In this study, however, intuitive eating was operationalised as two items adapted from the Tylka Intuitive Eating Scale: "I trust my body to tell me how much to eat" and "I stop eating when I am full". Among young adult men, 74.8% indicated that they trusted their body to tell them how much to eat, and 79.1% reported that they stopped eating when full. Among young adult women, 64.8% indicated that they trusted their body to tell them how much to eat, and 76.4% reported that they stopped eating when full [12]. Men consistently report higher levels of intuitive eating than do women [10]; this gender difference is particularly large for Caucasians [2].

## Feasibility of adopting intuitive eating at a population level

Several studies have examined the feasibility of adopting intuitive eating within the confines of clinical trials (e.g., [13–15]). Larkey [16] investigated dietitians' perceptions of barriers to the implementation of intuitive eating and Health at Every Size (an intuitive eating and mindfulness program) in community settings. No study we are aware of, however, has investigated the feasibility of adopting intuitive eating in the general population. In addition, little research exists regarding the extent to which the general public is aware of or understands intuitive eating. This is despite recent public interest in and exposure to body positivity, including the naming of body image activist Taryn Brumfitt as the 2023 Australian of the Year. Body positivity and intuitive eating share common elements, such as respecting one's body [17].

The aim of this study was to explore peoples' understanding of and reactions to intuitive eating, including perceived barriers and enablers to implementation, to provide a foundational understanding upon which public health policy around healthful eating can be built.

# Methods

## Study design and participants

Three focus group discussions were conducted with a total of 23 participants. Demographic characteristics of each group consisted of: (1) young women, aged 18–24, with no children; (2) women aged 35–45 with primary school aged children; and (3) men, aged 35–50, living with a partner and with pre- or primary school aged children. These three group demographics were selected to target significant age and life-stages in which shifts in eating behaviours may occur [18]. The groups were conducted in a hotel conference facility in Bendigo, a regional centre of Victoria, Australia, with participants recruited from Bendigo city and outlying areas. A non-metropolitan area was chosen for this research because rates of people with a BMI $> 30kg/m^2$ are generally higher in non-metropolitan areas in Australia [19].

The focus group discussions were guided by a Discussion Guide jointly developed by MM, NV, and ED. The Discussion Guide was not pilot tested. Repeat interviews were not carried out. The number of focus groups and participants was based on inclusion of an adequate representation of key demographic groups and budget, rather than data saturation. Therefore, findings should be considered preliminary.

## Focus group moderator and reflexivity

MM conducted the three focus groups. MM holds a Graduate Diploma in Applied Psychology and a Diploma of Applied Science (Naturopathy). He has 20 years' experience conducting qualitative research. At the time of the study, MM was Director of Qualitative Research at the Social Research Centre. MM is male and uses he/him pronouns. Prior to study commencement, MM had no relationship with the focus group participants. Participants knew nothing about the researcher, other than that he would be talking to them about their eating habits and how they made decisions around food. At the time of conducting the focus groups, MM's BMI was approximately 25–26 kg/m$^2$. He was using an intuitive eating approach some of the time.

## Procedure

Recruitment was conducted by a professional recruitment agency over the telephone from databases of people who had previously agreed to take part in market and social research. During the recruitment process, a screening questionnaire was used that included several questions about body weight and history of weight loss diets. To ensure a good mix of respondents for the discussions, the following quotas were set for each group: at least two in each group who had previously been on a weight loss diet and at least two who had never been on a weight loss diet; and at least three in each group who reported that they were "over my most healthy weight". Each participant provided oral informed consent before participation. Prior to the start of each focus group, MM provided a verbal explanation of the research and its purpose, and asked participants if they consented to having the discussion recorded and for the recordings to be used for the purposes of analysis and producing papers and reports. Participants were paid AUS$70. This approach was approved by the IRB.

The concept of intuitive eating was introduced to each of the groups following a discussion about diets and eating patterns. The term, intuitive eating, was first mentioned without any explanation and participants were prompted to discuss what they thought it might mean. This was followed by the provision of an explanation of intuitive eating, and a subsequent detailed discussion of the concept. The following definition of intuitive eating was provided:

> There are lots of ways to decide when, what, and how much to eat, such as because it's mealtime, the food looks really good, your friends are having it, out of habit, because you're following a diet, because you're hungry, etcetera.

> I'm going to explain an approach to eating called Intuitive Eating. The basic idea is that, if listened to, the body intuitively "knows" how much and what kinds of food to eat, both to maintain a healthy diet and an appropriate weight. This concept is sometimes referred to as "body wisdom". There are many things that work to override this innate body wisdom, such as diets, being made to clean one's plate as a child, eating because it's "dinner time", advertisements encouraging people to eat even when they're not hungry, and so on.

> The fundamental principles of intuitive eating are to regain body wisdom so that you eat only when you're hungry and stop eating when you're no longer hungry. There is also no restriction on the types of food you can eat (so, no "good" and "bad" foods) because the body will naturally choose a variety of foods that provide you with nutritional balance.

## Data collection

The focus groups and coding were conducted in 2010. Only MM and the participants were present during the discussions. With the permission of participants, all research sessions were

recorded and transcribed. The topic of intuitive eating was introduced first by name, followed by a definition. Participants were then asked what they thought about such an approach, and any barriers they saw in implementing such an approach. Each focus group lasted approximately 90 minutes. After each focus group, MM documented his initial reflections. Transcripts were not returned to participants for comment and/or correction.

### Data analysis

The transcriptions were thematically analysed by MM using an inductive descriptive approach [20] located within a constructionist perspective [21], following the general step-by-step approach outlined in Murphy et al [22]. An inductive descriptive approach was used to allow the data to drive the analysis, which was appropriate given the exploratory and descriptive purpose of the study. Themes were created by grouping codes with a central organising concept [23, 24]. A constructionist perspective assumes that multiple views of reality exist and attempts to understand the social construction of reality from the viewpoint of the study participants [25].

The analysis approach involved the following steps: (1) reading a sample of transcriptions to identify initial key themes and subthemes; (2) preparing a coding guide based on these themes and sub-themes; (3) reading/re-reading each transcript and coding against the guide, adjusting themes and sub-themes as necessary; (4) collating coded responses for each theme and subtheme. The results were discussed with the other co-authors and the first author also read the transcripts. All three authors agreed with the findings.

A selection of quotes has been included to illustrate themes; they do not necessarily reflect the specific attitudes of or differences between each demographic group. Each quote has been identified according to the group in which it was said. Participants did not provide feedback on the findings.

### Ethics

This project received ethics approval from Charles Sturt University (2010/144).

### Results

### Initial perceptions (prior to explanation)

Prior to participation in the focus groups, none of the participants had ever heard of intuitive eating. Initial reactions to the term were indicative of responses that might be expected if these participants were to come across such a reference in the public domain. In response to the term, the common perception was that intuitive eating was about eating what you feel like. Overall, participants believed that this was unlikely to be a good thing, as they believed that they were more likely to desire food that was not healthy (i.e., high in fat and sugar and based on looks and taste rather than nutrition) than food that was consistent with a healthy diet. Similarly, some were concerned that intuitive eating would be the same as responding to cravings, and based on their experience, cravings tended to be for foods that were not healthy.

Given these initial reactions, it was apparent that, without a more detailed explanation, intuitive eating was regarded as being similar to emotional eating, and unlikely to be perceived as eating healthily. It was apparent that participants did not really trust themselves to base food choices on what food they wanted or to make healthy choices intuitively.

*"If I tried that I'd die of chocolate poisoning in about three weeks."*

*(Men, 35–50)*

*"It could be dangerous. . . what you crave is quick fixes, quick anything fixes and that's what you get through sugar, so it can be dangerous, yes."*

*(Women, 35–45)*

Across the groups, a small number of participants had a reasonably accurate idea of what the term might mean, although even amongst these people there were doubts about the application of the concept.

*"I think, like intuition, and intuition to me is like, your body telling you what you should be eating, so I'd think maybe if you were low on calcium, that's the day you feel like a really big glass of cold milk, I think that's what it would be to me."*

*(Women, 18–24)*

*"Knowing what you need rather than what you want."*

*(Men, 35–50)*

*"Something along the lines of, um, eating. . .letting your body guide you to what you need to eat and when you need to eat it."*

*(Women, 35–45)*

Some participants in the group of men thought that the concept sounded boring, as they perceived it as suggesting that food be regarded primarily as fuel, rather than being valued for taste. One of the men was concerned that "intuitive" meant unsubstantiated by evidence. From this perspective, he believed that intuitive eating was likely to be another fad that had no scientific backing. Several other participants also felt that the concept sounded like a fad, suggesting the need for evidence of its effectiveness.

*"Intuition as a word, it's a belief system, without facts, it's the intuition, I just believe it, but I can't justify that belief system, so if you are going to go down that path, intuitive eating is, I believe it's useful for this purpose, but I really don't have the facts to back that up."*

*(Men, 35–50)*

*"It kind of still sounds like another fad thing though."*

*(Women, 18–24)*

## Response to explanation

Once the definition of intuitive eating was provided, participants were somewhat more interested in the concept, although they had substantial doubts about its efficacy.

*"It would be good if it worked."*

*(Women, 18–24)*

*"It makes sense. . . eat when you are hungry and don't eat when you are not. Hello!"*

*(Women, 18–24)*

*"I like the idea of just eating when you are hungry."*

*(Women, 35–45)*

Intuitive eating did appear to make inherent sense to most participants, but many also immediately stated that they believed that our lives are so far from being *lived* intuitively that *eating* intuitively did not seem realistic. Participants discussed several aspects of our modern existence that they believed potentially interrupted the capacity for intuition with respect to food. These included time demands–that so much of modern life needs to operate according to set times of the day and week. Another was availability of foods–that the choice of food products, including processed and manufactured foods, means that what people consume is not in itself intuitive. This was thought to be a far bigger lifestyle issue than just about eating, and would require substantive lifestyle changes. Finally, participants mentioned use of time, including daily and yearly cycles. In this context, two participants referred to intuitive eating as what they were more likely to do if they were camping, because then their eating patterns would be more in tune with daily cycles. Most participants had difficulty seeing how intuitive eating could be added into their lifestyles, but rather was regarded as requiring a significant change of lifestyle.

*"Like, it would be more a lifestyle thing than a diet. . .. it's not a short-term thing, it's a change [in lifestyle]."*

*(Women, 18–24)*

*"Unless you were living a very alternate lifestyle you're not going to change all the other influences."*

*(Women, 18–24)*

*"It would be if we went back to an idyllic rural lifestyle, you know if we went back to a situation where we were in touch with you know your sheep, your vegie patch. . . you go fishing, your hunting, your gathering."*

*(Men, 35–50)*

*"You can do that, but we live in, one of the reasons we have an obesity problem, is there's too much access, food is everywhere."*

*(Men, 35–50)*

*"It doesn't fit into western culture."*

*(Women, 35–45)*

In addition, some participants strongly disagreed with the notion that there was no such thing as good or bad food. Several participants referred to "junk foods" as a counter argument; some referred to specific food products or additives as being "non-food" and therefore "bad food".

*"I agree that everything. . .like the five food group, anything in moderation you can have oils, fat, protein, carb, anything in moderation, I agree with that, but I don't think that some of the foods that are available to us in the supermarket are any good for us at all. There's no point in having them; they don't have any nutrition, and they actually have things that harm your body." (Women, 35–45)*

*"Things that have got trans fats, I don't think we should be eating."*

*(Women, 35–45)*

Some participants believed that the "no good or bad food" aspect of the message would result in their putting on a lot of weight, because they would not be modifying their eating behaviours to restrict high fat or high sugar content foods in the way that they did at the moment.

*"Gives you a lot of leeway, so if you are wanting to lose weight, you are allowed to eat bad food in this?"*

*(Women, 18–24)*

*"I reckon I'd put on weight... yeah, the no good or bad foods, I'd be like, well if it it's not bad for me, so I can just eat it."*

*(Women, 18–24)*

One of the men, who throughout the discussion had talked about his diet in a way that indicated that he ate excessively and not in a very well-balanced manner, responded to the explanation of intuitive eating with the comment that this was his approach to eating–to eat whatever he felt like. He believed that the explanation was suggesting that his was an appropriate way to eat. These responses demonstrated that a significant amount of education would be needed for the concept to be properly understood. Several participants mentioned this need for education around food and intuitive eating.

*"You also need to have education, we've been exposed to so many of the bad foods, so we don't actually know the equivalent in the good foods, so if you knew that, oh I'm really craving, like chicken Twisties, but you could have roast chicken instead and it would satisfy the same craving."*

*(Women, 18–24)*

*"So you have to go back to the beginning and recondition. I think we don't know what food is. A lot of people don't know what food is."*

*(Women, 35–45)*

Moreover, while participants agreed that there was something interesting in the idea of intuitive eating, the impression of it as being unrealistic meant that they would be unlikely to spend a great deal of time or effort finding out more about it.

*"Ahm, I like the sound of it, but like you just said, [if I saw it as a heading in a magazine I would] read it and move on, I always flick through the magazines... but to be fair, it might just end up in the same pile of, 'not going to happen'."*

*(Women, 18–24)*

## Perceived barriers to eating intuitively

While some of the immediate responses presented above were barriers to the idea of intuitive eating, the notion of barriers was specifically probed and explored in the groups. Perceived barriers included food availability, overcoming existing food habits, difficulty differentiating between desire and cravings, removing temptations, and life structures and patterns that limit flexibility.

**Food availability.**   Some participants felt that a major barrier to adopting intuitive eating was that they did not have a sufficient range of foods available to make intuitive choices. They felt that they would only be able to follow their intuition if all foods were available all of the time. This was thought to be a particular problem for certain groups of people; for example, for people who lived in more rural or remote areas.

*"I don't think the right foods are available to a lot of people."*

*(Women, 35–45)*

**Overcoming existing food habits.**   A major issue for participants across all groups was the idea that they were already accustomed to certain foods and eating practices, and that any habits or pre-existing conditioning would interrupt the capacity for intuition. Some believed that they were already conditioned to eat certain things, and that it would take a long time to overcome their existing non-intuitive diets and approaches to eating.

*"It might not help you nutrition wise if you are. . . too busy eating your cheese burger or whatever, you might not really have the sense of mind to actually want the foods that are going to help you feel a bit more motivated or a bit healthier, or what not."*

*(Women, 18–24)*

*"Well, my body likes what I like."*

*(Women, 18–24) [in response to comment that intuitive eating sounds like just eating whatever you like]*

Related to this issue, some participants talked about having quite unhealthy approaches to food, including craving certain foods and having unhealthy eating patterns, and believed that unless these were addressed first, they would not be able to make intuitive choices.

*"Maybe there are addictions in place that we need to deal with first before we can do that to stop our bodies craving the wrong things, so we start craving."*

*(Women, 35–45)*

*"Yeah, elimination process, and then maybe when you're pure, you know, purified."*

*(Women, 35–45)*

**Difficulty differentiating between desire and cravings.**   It was apparent that a substantial limitation of the concept of intuitive eating was that participants were not sure how to distinguish between intuition, desire, and cravings. Some participants believed that eating what one wants and what one needs were mostly the same thing, but others were not as confident, and some felt they were very separate concepts. Across the groups, there was a strong and consistent tendency to believe that if they were eating what their body told them to, as per the intuitive eating definition, then they would eat cake and chips all the time. Some also felt that they would first need to re-educate themselves to not overeat.

*"You'd have to change your thinking. Like, if you've got a plate of food in front of you, you feel that you've got to eat the whole lot, not waste any."*

*(Women, 35–45)*

Some of the younger women were concerned that they, or their friends, would use these principles to justify starvation diets. They commented that some people they knew would excuse not eating by using words similar to those used to describe intuitive eating. From this perspective, they were concerned that the idea of intuitive eating could be abused and could allow people to justify unhealthy eating practices.

*"My housemate's sister, she does, she'll come down for tea but she hardly eats at all, she'll have like a McDonald's burger, one of those cheese burger things, she'll eat half of it and that will be all she has for the day, and then she'll eat a couple of lollies while she's at work and she says she's full. That's her constant line, but her doctor argues that she's turning anorexic because she's not eating enough, and that's her personal choice, but she does, everything on that list down to a T, including the no good or bad foods, but it doesn't really seem to be helping her necessarily."*

*(Women, 18–24)*

Some of the parents were concerned about how the concept could be applied to their children. They suggested that if they personally found it hard to understand that the concept was not just about eating whatever they felt like, they could not see how this could be applied to their children. Several participants discussed how this was relevant in terms of both the foods that their children ate and the times of their eating. In essence, they believed that it was their responsibility as parents to ensure that their children ate well, rather than to allow them to eat what they wanted.

*"Yeah, I mean look at it from the child's point of view, they know they don't want to eat the healthy thing, and if you talk about, I mean as you get older and you gain more experience and you understand more of what is good and why you should be eating the good, so if you went back to an intuitive, the body knows how much, and what foods to eat, well clearly it doesn't, if I give it to the children, if I'm going to let them pick, at that point in time, they should make the right choice more often, if it was truly intuitive."*

*(Men, 35–50)*

**Removing temptations.**   Many participants felt that, in order to be able to make intuitive choices, they would need to develop ways of resisting the temptations that they currently experienced. These included overcoming the impact of advertising and marketing of foods, and ensuring that they did not have easy access to the foods they believed they had cravings for (e.g., chocolate, sweets, etc.). This would require different approaches to shopping and the stocking of their refrigerators and pantries. Finally, they discussed the need to overcome what they regarded as normal reactions to foods in their immediate environment, including physiological responses to the sight and smell of other peoples' foods. One component of this issue would be working out how to deal with social events, such as eating out.

*"One place where it wouldn't work is the social eating, because if you're not hungry, and you go out, that's what I find, often if we go out for tea, I'm not actually very hungry, so I'll just pick at an entree or something, but you're still eating when you're not hungry, which is against what that is really about."*

*(Women, 18–24)*

*"Yeah you feel like you have to eat, because everyone else is eating, you are like, but I might see theirs and want something."*

*(Women, 18–24)*

*"Just the peer pressure, oh aren't you having anything?"*

*(Women, 18–24)*

**Life structures and patterns limit flexibility.** As indicated by people's initial reactions to the definition of intuitive eating, there was a strong and consistent belief that the practice of intuitive eating was not compatible with normal daily routines and structures, and would be difficult to maintain. This was thought to be especially problematic in terms of school and work, which often included set starting, finishing, and meal break times. Other activities, such as sports and social activities, were also commonly governed by set times. Participants explained that their food choices and timing needed to be organised around these, thus making a truly intuitive approach to eating impossible.

*"And you get your lunch break at work, if you are not eating then, what do you do?"*

*(Women, 18–24)*

*"Yeah you can't say oh sorry I can't go to that meeting at three o'clock, because I'm hungry."*

*(Women, 18–24)*

*"Oh well I, for example, tomorrow is hamburger Friday, we have hamburger Friday at work, so, if I don't feel like a hamburger tomorrow at lunch time, irrelevant, it is hamburger Friday!"*

*(Men, 35–50)*

Several of the parents, especially those with young children, reported that they believed that it was important to develop regularity in their children's habits. This included having set eating times. Some felt that it might be possible for adults to eat only when they were hungry, but that it was impractical for the routines of their children.

*"But it goes against everything we teach our kids, like I go out to homes and talk about the importance of routines and structure and everything like this and there's nothing about that, so as an adult you are putting that onto your children who don't have the discipline to do that. They've got the adults in their life to give them that discipline... How can you live by that and you have one set of morals that you teach the kids and do another it just wouldn't work; they need to mirror you and see you doing it?"*

*(Women, 18–24)*

*"Yeah that's like me and my kids, we put dinner on the table at six and that's when they eat."*

*(Women, 18–24)*

Participants discussed the idea that food choices, including what was eaten, how much was eaten, and when food was consumed, were not simply individual choices, especially within a family structure. Therefore, the question was raised about how intuitive eating would accommodate preparing food for a family.

*"I guess the other thing too is with that, you know, the whole eat when you are hungry, stop when you are not sort of thing, ahm, very impractical in a family environment, because you are not, you are cooking for the whole you know, you've just cooked dinner, or something for yourself and the kids* come in *and say oh, I'd like some of that, so you're back in there again cooking more or whatever."*

*(Men, 35–50)*

*"A lot of our structure goes around mealtime so it would actually. . .it would. . .how we break up our day and organise our time so it would make it difficult. . .the life skills that we're trying to teach our children are all based on, yeah, regularity."*

*(Women, 35–45)*

Some participants commented that they might be better able to eat intuitively when they did not experience external demands on their time or activities. Some thought that this could be possible on the weekends, especially amongst those who reported that they might have no demands from children in terms of time-based activities or nutrition needs.

*"On the weekend you might do that, you might be out gardening, mowing the lawns, visiting, whatever, haven't had lunch yet, shit, it's three o'clock. You just haven't had that need to eat, you've been busy or not hungry, so you just keep going until you are, and then you realise, oh it's three o'clock."*

*(Men, 35–50)*

## Perceived benefits and motivators to eating intuitively

Despite these perceived difficulties, most participants felt that if they were able to overcome the barriers and achieve a state of intuitive eating, there would be a range of benefits. Overall, the idea of living more in tune with one's body was considered appealing.

*"Yeah, I think it would be good. I think that it would be. . .you would feel like you were. . . in tune."*

*(Women, 35–45)*

Participants felt that, with an intuitive eating approach, they would eventually improve their food habits, with the main benefits that they would end up eating smaller portions and a healthier range of foods. As a consequence, they believed that they would eventually attain a better diet and healthier body weight. There was some understanding that it would take time for eating to become intuitive. These thoughts were premised on the idea documented previously, that people's existing food habits had developed patterns that were not intuitive, and that it would take some time to overcome these. Some appreciated that over time they would eat different foods and adopt different eating patterns, and that these approaches would eventually become their new pattern of eating.

*"But I reckon if you'd been doing that for two years, you would know, like your body clock would. . . form a pattern."*

*(Men, 35–50)*

*"You have to take it slow, we've all patterned now, and we are too old, if you had an empty slate and tried it, with lots of choice."*

*(Men, 35–50)*

*"Your body would get used to it, it would just kind of naturally want to pick the good foods. . . like you are consciously thinking about it, so you'd get more attuned to your own body and your own needs."*

*(Women, 18–24)*

**Additional information needed.** Participants were prompted to discuss any other information they would need before they considered adopting an intuitive eating approach. The main issue that people raised was a desire for evidence that intuitive eating "works" and that it is sustainable. They wanted to be sure that it was not just another unscientific diet fad, and that there was some research to demonstrate its value. They also wanted to know how it would affect people over time in terms of health and weight management.

*"You'd want to know some research on it and how it affects the body."*

*(Women, 35–45)*

*"And where is it coming from? Does it come from the Himalayas? When are you going to hand some stuff out?"*

*(Women, 35–45)*

*"And what do people look like who have done it?. . . Do they look lean and, um, good skin texture and all of that sort of stuff?"*

*(Women, 35–45)*

## Discussion

The research evidence is increasingly clear that intuitive eating may be helpful in supporting mental and physical health in the general population. Cross sectional surveys indicate that people who eat intuitively also have lower BMIs, and score higher on an array of physical and mental health indicators [1, 2, 7]. People with BMI >25kg/m$^2$ who complete intuitive eating programs tend to reduce or maintain their BMI, and improve on indicators of mental and physical health [1, 26, 27]. However, these findings are of little use from a public health perspective if the general population is unaware of intuitive eating, uninterested in becoming intuitive eaters, or deems such an approach too difficult or impractical to implement. It is therefore crucial to understand people's reactions to the concept of intuitive eating; yet, little research has been conducted on this topic. Erhardt [28], for example, states: "There is a distinct lack of qualitative research exploring people's experiences of learning to eat more intuitively".

Findings from this study suggest significant barriers to public adoption of an intuitive eating approach. The concept itself was unknown and generally misunderstood by focus group participants. None of the participants had ever previously heard of this approach to eating. Most thought it probably referred to eating whatever one felt like, which most associated with eating in response to cravings and/or eating unhealthy, "junk" foods. These findings suggest that significant public education would be necessary to clarity the concept of intuitive eating.

Somewhat heartening is research indicating that registered dietitians (in the United States) are increasingly knowledgeable about and favourably perceive intuitive eating [29]. This would suggest that people who seek dietary assistance may be informed about intuitive eating. However, a more recent experimental study of 31 registered dietitians found that, although most had heard of intuitive eating and believed they were incorporating some elements of intuitive eating into their practice, when presented with two case studies with identical medical indications, they recommended patients with higher BMIs lose weight in a manner that contradicted the main principles of intuitive eating [30].

Once intuitive eating was explained, most responses to implementing such an approach were negative. Participants felt that having complete choice around what they ate was unlikely to equate to a healthy or balanced diet, at least in the short term. They also argued that because everyday life was not intuitive in its structures, it would be difficult to eat intuitively. Switching to intuitive eating, therefore, was thought to require an entire lifestyle change, rather than just a change in food choices and eating patterns. They also argued that existing food patterns and preferences would need to be addressed before an intuitive approach to eating could be adopted. This would require addressing cravings and factors that affected taste preferences. In addition, participants discussed the fact that external factors outside the control of the individual influenced food choices, including advertising and marketing of food products and the availability of foods. Moreover, various social structures required set times (e.g., work, school, sports, and other social activities) and that food choices and eating times needed to be coordinated around these, which participants believed would make intuitive eating impractical if not impossible. Finally, parents were concerned about how the concept could be applied to children, as some believed that children required set time structures for meals and did not have the knowledge or awareness to make appropriate food choices.

Some of these same concerns are reflected in the literature from small intuitive eating trials. A qualitative study with 11 middle-aged women learning to eat intuitively, for example, found that the women were able to learn to eat more intuitively. However, they encountered social and environmental barriers, with the 'unconditional permission to eat' aspect of intuitive eating the most challenging [23]. A U.S. study with non-Hispanic black men with Type 2 diabetes identified multiple barriers to healthy eating (not specifically intuitive eating), including hard-to-break habits, limited resources and availability of food at home and in neighbourhood grocery stores, and perceived poor communication with health care professionals [31].

Despite these difficulties, participants appreciated that if they were able to overcome the various barriers and achieve a state of intuitive eating, then they anticipated a range of long-term benefits in regard to health and weight management. The fact that many participants expressed a desire to be provided with evidence that intuitive eating "works" suggests that, if provided with this evidence, they might be convinced to try to change their eating behaviours. However, participants also wanted evidence regarding how eating intuitively might affect people over time in terms of health and weight management. Most of the existing research, however, consists of either a single point in time measure, or follows people for only a short period of time–usually for between six months and two years. This last finding supports previous recommendations regarding the need to conduct research with longer-term outcomes [1].

If intuitive eating were to become a viable public health approach, this research suggests that people would need assistance in overcoming their existing habits and food preferences, and developing better insight into their bodies' needs, desires, and cravings. The structural barriers, however, would likely continue to be substantial. People would need to learn how to adapt the concept of intuitive eating to fit in with their work, family, and social schedules. As with many behavioural changes, eating intuitively would likely require a gradual shift in

behaviours and attitudes. A starting point would be implementation of a public health campaign educating people about intuitive eating and its connections with better health. Another step would be specific training of dietitians, GPs, and other medical and allied health professionals in intuitive eating and its potential benefits to their clients. Access to such services must also be examined, unless we want to further exacerbate the health divide been the rich and poor.

Most significantly, however, are the changes that must be made to the food environment if intuitive eating is ever to become a real option for most people. Intuitive eating requires choice of food, yet ample evidence demonstrates that true choice is often not available, and that more healthful and less healthful foods do not exist on an even playing field [32]. Supermarkets and other food retail outlets, for example, preference highly processed foods over fresh foods in their placement and pricing [33–35]. Many people, and particularly those living in lower socio-economic areas, live in "food deserts" and/or "food swamps"—areas with no access to a nearby supermarket or with a high ratio of fast-food outlets to fresh food outlets [36, 37]. Advertising spending favours highly processed foods over fresh foods by a large margin [38]. And the cost of fresh food has risen more rapidly than the cost of highly processed food [39–41].

## Study strengths and limitations

This study is the first we are aware of to explore the concept of intuitive eating with a general population. Its focus on people living in a regional town is an important feature given that research is often conducted in major cities, which can have a very different food environment and lifestyle. This study was conducted with 23 people from three demographic groups from a small area in regional Victoria, Australia. Additional research with larger numbers and other types of people, both within and outside of Australia, would be useful to understand the extent to which these findings hold more broadly. Identified barriers and enablers to uptake of intuitive eating are likely to be similar in other western populations.

The definition of intuitive eating provided to participants included the statement: "The basic idea [of intuitive eating] is that, if listened to, the body intuitively 'knows' how much and what kinds of food to eat, both to maintain a healthy diet and an appropriate weight." Participants may have interpreted 'an appropriate weight' as a lower body weight according to societal norms. A single researcher conducted the focus groups and analysed the data. With thematic analysis, coding quality is not dependent on multiple coders [42]. The results were discussed with the other co-authors and the first author also read the transcripts. All three authors agreed with the findings.

Finally, the focus groups were conducted in 2010 and we cannot be confident that the findings would be identical now. In particular, these findings do not reflect the potential impact on attitudes towards intuitive eating and eating choices of the COVID pandemic and associated lockdowns.

## Conclusion

Intuitive eating has great potential for improving people's physical and mental health. However, any public health approach must consider the realities of people's lives, including both personal and structural barriers.

## Acknowledgments

We would like to acknowledge the focus group participants, who generously shared information and insights about themselves and their families.

## Author Contributions

**Conceptualization:** Nina Van Dyke.

**Funding acquisition:** Eric J. Drinkwater.

**Investigation:** Michael Murphy.

**Methodology:** Nina Van Dyke.

**Writing – original draft:** Nina Van Dyke, Michael Murphy.

**Writing – review & editing:** Eric J. Drinkwater.

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
