## [Decision Letter · Decision Letter 0]

10 Feb 2023

PONE-D-22-32682What do people think of intuitive eating?  A qualitative exploration with rural AustraliansPLOS ONE

Dear Dr. Van Dyke,

Thank you for submitting your manuscript to PLOS ONE. After careful consideration, we feel that it has merit but does not fully meet PLOS ONE’s publication criteria as it currently stands. Therefore, we invite you to submit a revised version of the manuscript that addresses the points raised during the review process.

I have now received reviews on your manuscript. Both reviewers are positive about the eventual contribution of your work. They note the strengths and limitations (e.g., the age of the data) of your approach to understanding of intuitive eating. You will see that the comments made are complementary, representing the reviewers backgrounds. I would like to invite you to revise your MS according to these comments and resubmit, as per the instructions attached.BestSO

We look forward to receiving your revised manuscript.

Kind regards,

Stefano Occhipinti

Academic Editor

PLOS ONE

Journal Requirements:

2. In the ethics statement in the Methods, you have specified that verbal consent was obtained. Please provide additional details regarding how this consent was documented and witnessed, and state whether this was approved by the IRB.

Reviewers' comments:

Reviewer's Responses to Questions

**Comments to the Author**

1. Is the manuscript technically sound, and do the data support the conclusions?

Reviewer #1: Yes

Reviewer #2: Yes

2. Has the statistical analysis been performed appropriately and rigorously? 

Reviewer #1: N/A

Reviewer #2: N/A

3. Have the authors made all data underlying the findings in their manuscript fully available?

Reviewer #1: Yes

Reviewer #2: Yes

4. Is the manuscript presented in an intelligible fashion and written in standard English?

Reviewer #1: Yes

Reviewer #2: Yes

5. Review Comments to the Author

Reviewer #1: This manuscript reports on a study involving 3 focus groups examining people’s understanding of the concept of ‘intuitive eating’. Two of the focus groups were made up of women (younger and older age groups) and one of men (older age group). The findings show that all participants were not previously familiar with the concept, and had a number of reservations about it once is was explained to them. The findings highlight challenges that could be faced when attempting to roll out intuitive eating interventions as part of a public health strategy. The manuscript is generally easy to follow and explores a gap in the research on an interesting topic. The findings could be more clearly tied to what is already known, as well as how they can inform future directions. A limitation is that only one of the 3 authors both conducted the focus groups and conducted the thematic analysis. There are a few areas where the writing needs tightening up. The data has been made available, however, the authors should make sure that the transcripts being made available are compliant with their ethical approvals. It seems to me that they are potentially identifiable (they appear to contain First names, and other information that may allow others identify them). This would need to be confirmed prior to publication taking place.

Major issues

1. Transcripts have not been anonymised. It’s great that the authors have made their transcripts available. However, there are issues around confidentiality for participants, as names and other potentially identifying details appear to still be in the transcripts. It is also important that the authors confirm they acquired explicit participant consent for the transcripts to be shared in such a broad way. The authors should check and confirm there are no confidentiality issues arising from sharing the transcripts.

2. Coding of the data. The authors do note this as a limitation on p23. It appears that only one of the 3 authors was involved in coding the data. Were the results discussed with other co-authors and refinement undertaken? Did other co-authors read the transcripts and are in broad agreement with the findings?

3. There are a few places in the Discussion where the study findings are followed by statements about existing research, but there aren’t sentence(s) to explain how the findings link to the existing research. For example, the manuscript notes that significant public education is required, but the previous research outlined in subsequent sentences is about dietitians knowledge of intuitive eating (p 20, lines 294-502). The link isn’t clear to me. Are the authors suggesting that dietitians will be the ones educating the public?

4. The Introduction flows logically and introduces the reader to the topic area. I would have liked to see a more detailed discussion of how intuitive eating is operationalised (lines 50-52, p3), and the Tylka Intuitive Eating Scale (or at least the relevance of the scale cut-offs being provided in lines 91 and 93 on p4).

Other comments

5. The manuscript is easy to follow and is logically arranged. At times the language used verges on the colloquial – I would recommend a read through and tightening up of some language/sentences. Occasionally a sentence contains multiple ideas and it would be clearer to split into two.

Abstract

6. Conclusion could be tighter and focus more on the specific findings of the manuscript.

Introduction

7. Important to not give impression that this is a feasibility study (line 106 states no study has investigated feasibility in the general population, and then line 108 says aim of study was to fill this gap)

Methods

8. Line 113 p5, the authors state the discussion adheres to “Tong et al. (2007)” checklist, but this reference does not seem to be in the reference list, and this is not the journal reference style

9. Line 140 p6 states participants were paid an incentive ‘according to current market rates’. I assume the authors mean current at the time the study was conducted? It’s unclear to me how much participants would actually have been paid.

10. Include year of data collection in ‘Data Collection’ section

11. Which institutional review board or ethics committee provided the ethics approval?

12. I would have liked to see a justification of why no young men were included in the focus groups, as there are two groups of women and only one group of men (older). Especially as there appears to be some research suggesting men are more likely to engage in intuitive eating.

Results

13. The authors often highlight when male participants hold a particular view (e.g., line 221 p 10) but it wasn’t clear when views were held only by women – although there are a number of themes where the quotes are only from women. Should the reader assume that means gender made a difference, or is it just that these were better illustrative quotes. I suspect this could be related to there being a greater number of female participants?

14. Line 286-287, p 12. This sentence read to me as if it was an author interpretation, rather than something a view expressed by participants, but then when I read the illustrative quotes lines 288-293 the participants themselves suggest that education would be required. I think the authors should make more of this as it is a stronger argument.

Discussion

15. I would have liked to see the Limitations (p23) more clearly (briefly) note why the limitations identified are potentially problematic.

16. I also wonder if the authors are doing themselves a disservice by focusing on generalisability in the Limitations – it’s not always an expectation that qualitative results will apply to the general population. Also, focusing on the views of people living in a regional town could be considered an important feature, given that research is so often conducted in major cities, which can have very different food landscapes.

Reviewer #2: Thank you for the opportunity to provide a first-round review of this manuscript. It describes a qualitative study and reports the findings from 3 focus groups conducted in 2010 where the intuitive eating concept was introduced to participants. An inductive approach was taken to understand the barriers and facilitators to participants adopting intuitive eating. The title reflects the content of the manuscript well. The writing style is uncomplicated and readable for a non-specialist audience. The quotes used illustrate the themes well. There are some weaker conceptual and methodological elements that I am confident can be overcome with a major revision. Overall the work is interesting and contributes to our understanding of important elements to consider when designing public health campaigns and health promotion endeavours which aim to support intuitive eating styles for Australian adults.

The following elements require revision/reflection in order to strengthen the manuscript:

Data was collected in 2010. Since that time there has been significant growth in the academic understanding and definition of intuitive eating, as well as public interest and exposure to body positivity, of which intuitive eating is a common element. The age of the data is not necessarily a problem but the background should provide a more robust historical context into which the findings can then be positioned.

Methodological issues: with one person conducting the focus groups, devising the coding scheme and analysing the data, there is a risk of insufficient rigour. There are another two authors on the paper, did they conduct any review of the analysis documents or methods, even if retrospectively?

It is acknowledged in the manuscript that analysis was deemed complete when the transcripts had been analysed rather than on the basis of data saturation (the constraints were well described) - did the author have any sense of whether saturation had been achieved for some themes or if it would have been achieved with another focus group (or many other focus groups)? If saturation was not seemingly imminent, the findings may better be described as preliminary.

How was the definition of intuitive eating used in the focus groups developed? Is there a reason that it differs from academic definitions available at the time of the data collection? It’s quite different from the ten intuitive eating principles of the original authors, and doesn’t include the concept of ‘gentle nutrition’ which aligns with body-food congruence element of the IES. The closely-related concepts of mindful eating and attentive eating are also not mentioned but should be.

Grounded Theory methodology would stipulate that analysis commences as soon as the first data was collected but it is not clear whether this was done. The age of the data collected is an issue – need to clarify whether analysis was conducted concurrently with data collection as per grounded theory protocol and informed by the insights of the researcher at the time. If analysis was conducted more recently, the historical context may be lost and both the more advanced understanding of the researcher (over the past 13 years).

Similarly, with the use of Grounded Theory methodology, deeper connections between identified themes would be expected in order to generate a coherent theory of findings. For example, identifying body distrust and fears of weight gain as common threads arising from the proposition that body cues are able to regulate eating behaviour could have been used to develop a unified theory of responses. The manuscript provides a good descriptive treatment of the data but does not adequately reflect grounded theory per se. My recommendation is that rather than describing it as Grounded Theory, it is described as inductive descriptive.

Throughout the work care should be taken to differentiate the psychometric construct of flexible dietary restraint as measured by the Intuitive Eating Scales 1 & 2 (a construct that can be measured in every human and does not rely on people ‘knowing’ about ‘intuitive eating’) from the concept of Intuitive Eating (a set of eating and body acceptance principles that someone may intentionally attempt to adopt). Because the Intuitive Eating Scales are inverted dietary restraint measures, cross-sectional studies of intuitive eating join a large research field in eating behaviour relating to dietary restraint and should not be presented in isolation.

Accordingly, a more in-depth description of the Intuitive Eating concept is warranted in the background. At the moment only the 4 factors in the intuitive eating scale are named, without an explanation for the reader. Put another way, if the definition of intuitive eating as reflected in the intuitive eating scales was used, there should be attention given to the use of strategies other than food to soothe emotional states, that unconditional permission to eat relates to unconditional body size/weight/shape acceptance, responding to spontaneous urges to consume nutritious foods and responsiveness to body cues of hunger and fullness. Alternatively, if the definition of Intuitive Eating is to be derived from the works of Tribole and Resch (as stated initially in the background), and which most of the intervention studies have used, then the ten Intuitive Eating Principles should be used. At a minimum, these aspects of competing definitions should be foregrounded in the background section of the manuscript. They are touched on in the discussion so I'm confident that the authors are aware of the issues.

It’s really interesting to essentially ‘brand test’ the term ‘intuitive eating’ and that it was assumed to be synonymous with emotional eating, which participants seemed to interpret as negative. It shows that intuitive eating isn’t going to be self-explanatory with only limited information and support provided.

I concur with the authors that their findings show that the concept of intuitive eating is not so 'intuitive' to understand by community members, but also see an opportunity for the authors to place this conceptual ambiguity in context as a reflection or echo of the academic ambiguity around intuitive eating in the literature. These themes could be developed in the discussion section for greater impact on the field.

Detailed feedback:

Line 43, the first sentence is unnecessary.

Line 55 and elsewhere: The terms ‘overweight’ and ‘obese’ are considered stigmatising as they pathologise higher body weight in the absence of unified pathophysiology. Recommend stating the BMI cut-offs used in the research being discussed, eg women with a BMI >25kg/m2, or if no specific cut-offs were used, refer to this population as ‘larger-bodied’ or ‘much larger-bodied’ if referring to populations with markedly high BMIs.

Line 76: clarify that you mean ‘anti-obesity’ rather than ‘obesity’

Line 89: there was a representative sampling of New Zealand

Line 124: r’ates of people with a BMI>30kg/m2’ or ‘higher body weight is more prevalent’

Line 129: MM’s own relationship with his body and whether he perceived that he used eating behaviours to control his body size is relevant. Did his body mass exceed BMI25kg/m2 at the time of the interviews? Did he personally ascribe to intuitive eating? It is not possible to be bias-free in this context. His body size may also have influenced the participant’s responses.

Line 154: was the coding guide and theme identification corroborated by another researcher? Why, why not?

Line 174-191: this relates to focus group protocol and should be located within methodology rather than results.

Line 184 – Intuitive Eating does not claim to be a weight management strategy or effective at reaching an ‘appropriate weight’ – which participants might have interpreted as lower BMI as per social norms. Limitations should state that this could have mislead participants.

Line 543: ‘be’ should be ‘with’ or similar

6. PLOS authors have the option to publish the peer review history of their article (what does this mean?). If published, this will include your full peer review and any attached files.

Reviewer #1: No

Reviewer #2: **Yes: **Dr Fiona Willer, AdvAPD, PhD

---

## [Author Response · Author response to Decision Letter 0]

31 Mar 2023

Response to Reviewers

What do people think of intuitive eating? A qualitative exploration with rural Australians

Thank you for the opportunity to revise and resubmit this manuscript. We appreciate the time taken by the Reviewers to review this paper and for their helpful comments.

Editor’s Comments:

Authors’ response:

The manuscript now meets PLOS ONE’s style requirements according to the “PLOS ONE Title, Author, Affiliation Formatting Guidelines” and file naming.

2. In the ethics statement in the Methods, you have specified that verbal consent was obtained. Please provide additional details regarding how this consent was documented and witnessed, and state whether this was approved by the IRB.

Authors’ response: 

This information has been added to the Methods section.

Reviewer: 1

Reviewer #1: This manuscript reports on a study involving 3 focus groups examining people’s understanding of the concept of ‘intuitive eating’. Two of the focus groups were made up of women (younger and older age groups) and one of men (older age group). The findings show that all participants were not previously familiar with the concept, and had a number of reservations about it once is was explained to them. The findings highlight challenges that could be faced when attempting to roll out intuitive eating interventions as part of a public health strategy. The manuscript is generally easy to follow and explores a gap in the research on an interesting topic. The findings could be more clearly tied to what is already known, as well as how they can inform future directions. A limitation is that only one of the 3 authors both conducted the focus groups and conducted the thematic analysis. There are a few areas where the writing needs tightening up. The data has been made available, however, the authors should make sure that the transcripts being made available are compliant with their ethical approvals. It seems to me that they are potentially identifiable (they appear to contain First names, and other information that may allow others identify them). This would need to be confirmed prior to publication taking place.

Major issues

1. Transcripts have not been anonymised. It’s great that the authors have made their transcripts available. However, there are issues around confidentiality for participants, as names and other potentially identifying details appear to still be in the transcripts. It is also important that the authors confirm they acquired explicit participant consent for the transcripts to be shared in such a broad way. The authors should check and confirm there are no confidentiality issues arising from sharing the transcripts.

Authors’ response: 

We thank the reviewer for this feedback(!) After careful consideration, we have removed the transcripts from Figshare and have included the following data availability statement with our revised submission:

De-identified transcripts are available upon request.

Due to the nature of the data (i.e. a small number of focus group participants from a single geographic area), it is very difficult to anonymize the data. In addition, the participants did not provide explicit consent for the transcripts to be shared publicly.

We have gone back through the transcripts and deleted any identifying information. Should there be a request for the transcripts, we will first clarify with the HREC that there are no confidentiality issues arising from sharing these transcripts.

2. Coding of the data. The authors do note this as a limitation on p23. It appears that only one of the 3 authors was involved in coding the data. Were the results discussed with other co-authors and refinement undertaken? Did other co-authors read the transcripts and are in broad agreement with the findings?

Authors’ response: 

We have added the following statement to the Limitations section and added a reference: 

With thematic analysis, coding quality is not dependent on multiple coders (41). The results were discussed with the other co-authors and the first author also read the transcripts. All three authors agreed with the findings.

41. Bruan, V. & Clarke, V. (2022). Conceptual and design thinking for thematic analysis. Qualitative Psychology, 9(1), 3-26.

3. There are a few places in the Discussion where the study findings are followed by statements about existing research, but there aren’t sentence(s) to explain how the findings link to the existing research. For example, the manuscript notes that significant public education is required, but the previous research outlined in subsequent sentences is about dietitians knowledge of intuitive eating (p 20, lines 294-502). The link isn’t clear to me. Are the authors suggesting that dietitians will be the ones educating the public?

Authors’ response: 

We have added a sentence that hopefully clarifies the link between the two sentences. The first point was that the findings suggest that most people in the general population have not heard of intuitive eating or misunderstand what it is. The second point was that perhaps a subset of the population – those who have seen a dietitian – may have learned about intuitive eating through their practitioner. 

We have also gone back through the Discussion section to ensure that any statement about findings followed by the existing research are clearly linked.

4. The Introduction flows logically and introduces the reader to the topic area. I would have liked to see a more detailed discussion of how intuitive eating is operationalised (lines 50-52, p3), and the Tylka Intuitive Eating Scale (or at least the relevance of the scale cut-offs being provided in lines 91 and 93 on p4).

Authors’ response: 

We have provided additional information in the Introduction about how intuitive eating has been operationalised.

In our 2022 paper on intuitive eating amongst rural Australians (citation 10), we found it helpful to provide figures around the % of that population that “eats intuitively”, although of course intuitive eating is not binary. In an email exchange, Tylka suggested that an average score of 4.0 or higher could be considered “intuitive eating”. We have clarified that this is an average of 4.0 out of 5.0.

Other comments

5. The manuscript is easy to follow and is logically arranged. At times the language used verges on the colloquial – I would recommend a read through and tightening up of some language/sentences. Occasionally a sentence contains multiple ideas and it would be clearer to split into two.

Authors’ response: 

We have reviewed the manuscript with this suggestion in mind and made changes where appropriate. In particular, we have split some sentences.

Abstract

6. Conclusion could be tighter and focus more on the specific findings of the manuscript.

Authors’ response: 

The conclusion in the abstract has been rewritten to focus more tightly on the findings of the manuscript.

Introduction

7. Important to not give impression that this is a feasibility study (line 106 states no study has investigated feasibility in the general population, and then line 108 says aim of study was to fill this gap)

Authors’ response: 

Thanks for pointing this out. We have removed “fill this gap in the literature” as this phrase is confusing and unnecessary.

Methods

8. Line 113 p5, the authors state the discussion adheres to “Tong et al. (2007)” checklist, but this reference does not seem to be in the reference list, and this is not the journal reference style

Authors’ response: 

We have deleted this sentence as it is not needed.

9. Line 140 p6 states participants were paid an incentive ‘according to current market rates’. I assume the authors mean current at the time the study was conducted? It’s unclear to me how much participants would actually have been paid.

Authors’ response: 

AUS$70. We have added this information to the manuscript and moved it to the end of the Procedures section.

10. Include year of data collection in ‘Data Collection’ section

Authors’ response: 

This information has been added.

11. Which institutional review board or ethics committee provided the ethics approval?

Authors’ response: 

This information is included under “Ethics”.

12. I would have liked to see a justification of why no young men were included in the focus groups, as there are two groups of women and only one group of men (older). Especially as there appears to be some research suggesting men are more likely to engage in intuitive eating.

Authors’ response: 

The three demographics were chosen based on those groups thought most likely to have veered from intuitive eating – younger women (concerned with weight and body image), and women and men with children (having to fit eating around family activities, multiple food preferences, etc.) To make this point clearer, we have deleted “provide something of a cross-section of the adult population” as that wasn’t really a major aim.

Results

13. The authors often highlight when male participants hold a particular view (e.g., line 221 p 10) but it wasn’t clear when views were held only by women – although there are a number of themes where the quotes are only from women. Should the reader assume that means gender made a difference, or is it just that these were better illustrative quotes. I suspect this could be related to there being a greater number of female participants?

Authors’ response: 

Where only one gender held a particular opinion, this is specified in the paper. Thus, only (a sub-group of the) men thought that the concept of intuitive eating sounded boring (line 221). Elsewhere, opinions were held by both genders, regardless of whether only women (or men) were quoted.

14. Line 286-287, p 12. This sentence read to me as if it was an author interpretation, rather than something a view expressed by participants, but then when I read the illustrative quotes lines 288-293 the participants themselves suggest that education would be required. I think the authors should make more of this as it is a stronger argument.

Authors’ response: 

This is a really good point – thanks. We have added a sentence to make this clear.

Discussion

15. I would have liked to see the Limitations (p23) more clearly (briefly) note why the limitations identified are potentially problematic.

Authors’ response: 

We have revised this section of the paper and included this information.

16. I also wonder if the authors are doing themselves a disservice by focusing on generalisability in the Limitations – it’s not always an expectation that qualitative results will apply to the general population. Also, focusing on the views of people living in a regional town could be considered an important feature, given that research is so often conducted in major cities, which can have very different food landscapes. 

Authors’ response:

Thank you for these comments. And we agree. We have rewritten this section of the paper. I hope you don’t mind that we have used your language re: the focus on a non-metropolitan population. 

Reviewer #2: 

Thank you for the opportunity to provide a first-round review of this manuscript. It describes a qualitative study and reports the findings from 3 focus groups conducted in 2010 where the intuitive eating concept was introduced to participants. An inductive approach was taken to understand the barriers and facilitators to participants adopting intuitive eating. The title reflects the content of the manuscript well. The writing style is uncomplicated and readable for a non-specialist audience. The quotes used illustrate the themes well. There are some weaker conceptual and methodological elements that I am confident can be overcome with a major revision. Overall the work is interesting and contributes to our understanding of important elements to consider when designing public health campaigns and health promotion endeavours which aim to support intuitive eating styles for Australian adults.

Authors’ response:

Thank you

The following elements require revision/reflection in order to strengthen the manuscript:

Data was collected in 2010. Since that time there has been significant growth in the academic understanding and definition of intuitive eating, as well as public interest and exposure to body positivity, of which intuitive eating is a common element. The age of the data is not necessarily a problem but the background should provide a more robust historical context into which the findings can then be positioned.

Authors’ response:

The background section does include a discussion of more recent scholarship. For example, the start of the Background section discusses the gradual shift in the intuitive eating literature from a focus on reducing obesity to a focus on increasing mental health and wellbeing. It also mentions more recent evidence that is beginning to establish causal direction. We agree that there may be greater awareness amongst the public of concepts like body positivity, particularly in the past few months with the naming of Taryn Burmfitt as 2023 Australian of the Year. It is unknown, however, the extent to which intuitive eating is associated with a concept such as body positivity. We have added additional text under Feasibility of adopting intuitive eating at a population level.

Methodological issues: with one person conducting the focus groups, devising the coding scheme and analysing the data, there is a risk of insufficient rigour. There are another two authors on the paper, did they conduct any review of the analysis documents or methods, even if retrospectively?

Authors’ response:

We have added additional details to both the Analysis section and the Limitations section.

It is acknowledged in the manuscript that analysis was deemed complete when the transcripts had been analysed rather than on the basis of data saturation (the constraints were well described) - did the author have any sense of whether saturation had been achieved for some themes or if it would have been achieved with another focus group (or many other focus groups)? If saturation was not seemingly imminent, the findings may better be described as preliminary.

Authors’ response:

This is a fair point. We doubt that saturation was reached with only three groups, each representing a different age and gender demographic. Although we discuss size as a limitation, we agree that it would be most accurate to describe the findings as preliminary. We have added this statement to the end of the Study Design and Participants section. 

How was the definition of intuitive eating used in the focus groups developed? Is there a reason that it differs from academic definitions available at the time of the data collection? It’s quite different from the ten intuitive eating principles of the original authors, and doesn’t include the concept of ‘gentle nutrition’ which aligns with body-food congruence element of the IES. The closely-related concepts of mindful eating and attentive eating are also not mentioned but should be.

Authors’ response:

The definition of intuitive eating provided to participants was based primarily on the work of Tribole and Resch (1995) and Tylka (2006), and was intended to be easy to understand when read out to participants. We would argue that the reference to ‘maintain a healthy diet’ corresponds to ‘gentle nutrition’. As the focus was specifically on intuitive eating, rather than the closely-related concepts of mindful eating and attentive eating, these constructs were not explicitly included in the definition. However, we would argue that the references to ‘if listened to’ and ‘body wisdom’ touch on these constructs. 

Grounded Theory methodology would stipulate that analysis commences as soon as the first data was collected but it is not clear whether this was done. The age of the data collected is an issue – need to clarify whether analysis was conducted concurrently with data collection as per grounded theory protocol and informed by the insights of the researcher at the time. If analysis was conducted more recently, the historical context may be lost and both the more advanced understanding of the researcher (over the past 13 years).

Authors’ response:

Analysis was conducted at the time of data collection. We would argue that the grounded theory approach means that the analysis is happening even before data collection -- during the discussion groups -- as a critical element of analysis goes on when the moderator is choosing which questions to follow up on, which issues to probe in depth, and which to leave (Rossiter, J. R. (2009). Qualitative marketing research: theory and practice. Australasian Journal of Market and Social Research, 17(1), 7.)

Similarly, with the use of Grounded Theory methodology, deeper connections between identified themes would be expected in order to generate a coherent theory of findings. For example, identifying body distrust and fears of weight gain as common threads arising from the proposition that body cues are able to regulate eating behaviour could have been used to develop a unified theory of responses. The manuscript provides a good descriptive treatment of the data but does not adequately reflect grounded theory per se. My recommendation is that rather than describing it as Grounded Theory, it is described as inductive descriptive.

Authors’ response:

That’s a fair comment. We have made this change.

Throughout the work, care should be taken to differentiate the psychometric construct of flexible dietary restraint as measured by the Intuitive Eating Scales 1 & 2 (a construct that can be measured in every human and does not rely on people ‘knowing’ about ‘intuitive eating’) from the concept of Intuitive Eating (a set of eating and body acceptance principles that someone may intentionally attempt to adopt). Because the Intuitive Eating Scales are inverted dietary restraint measures, cross-sectional studies of intuitive eating join a large research field in eating behaviour relating to dietary restraint and should not be presented in isolation.

Accordingly, a more in-depth description of the Intuitive Eating concept is warranted in the background. At the moment only the 4 factors in the intuitive eating scale are named, without an explanation for the reader. Put another way, if the definition of intuitive eating as reflected in the intuitive eating scales was used, there should be attention given to the use of strategies other than food to soothe emotional states, that unconditional permission to eat relates to unconditional body size/weight/shape acceptance, responding to spontaneous urges to consume nutritious foods and responsiveness to body cues of hunger and fullness. Alternatively, if the definition of Intuitive Eating is to be derived from the works of Tribole and Resch (as stated initially in the background), and which most of the intervention studies have used, then the ten Intuitive Eating Principles should be used. At a minimum, these aspects of competing definitions should be foregrounded in the background section of the manuscript. They are touched on in the discussion so I'm confident that the authors are aware of the issues.

Authors’ response:

This is an interesting point. Although Tylka (2006; 2013) discussed the development of the IES as based on Tribole and Resche’s (1995) work, the IES does not include all 10 of Tribole and Resche’s principals – e.g. ‘movement – feel the difference’. The IES-1 did not include gentle nutrition, although this construct was added in the IES-2. Most of the empirical work on intuitive eating has used the IES-1 or IES-2 to measure intuitive eating.

The definition of intuitive eating provided to participants was based primarily on Tribole and Resch (1995) and Tylka (2006). We attempted to provide a reasonably broad definition that at the same time would be comprehensible to participants (particularly as they were read the definition).

We have added some additional text to the Introduction section to more clearly specify the 10- principals of Tribole and Resch (1995), as well as the three sub-scales of the IES-1 and a reference to the refinement to the scale in 2013.

It’s really interesting to essentially ‘brand test’ the term ‘intuitive eating’ and that it was assumed to be synonymous with emotional eating, which participants seemed to interpret as negative. It shows that intuitive eating isn’t going to be self-explanatory with only limited information and support provided.

Authors’ response:

Agree. We actually had a government wellbeing agency say they were not interested in supporting the concept of intuitive eating – mainly, as it turned out, due to misconceptions about what the term means, and also feeling that such an approach is not suitable with regards to children’s eating. It may be that another term is needed. 

I concur with the authors that their findings show that the concept of intuitive eating is not so 'intuitive' to understand by community members, but also see an opportunity for the authors to place this conceptual ambiguity in context as a reflection or echo of the academic ambiguity around intuitive eating in the literature. These themes could be developed in the discussion section for greater impact on the field.

Authors’ response:

This is also an interesting point. After much consideration, and given our approach to developing the definition of intuitive eating provided to participants, we felt that such a discussion was out of scope for this study and would detract from its focus.

Detailed feedback:

Line 43, the first sentence is unnecessary.

Authors’ response:

Agreed. This sentence has been deleted.

Line 55 and elsewhere: The terms ‘overweight’ and ‘obese’ are considered stigmatising as they pathologise higher body weight in the absence of unified pathophysiology. Recommend stating the BMI cut-offs used in the research being discussed, eg women with a BMI >25kg/m2, or if no specific cut-offs were used, refer to this population as ‘larger-bodied’ or ‘much larger-bodied’ if referring to populations with markedly high BMIs.

Authors’ response:

This is an interesting point. Last year, NV attended the International Conference on Obesity where, much to my relief, most of the conversations and papers in the public health stream (at least, the ones I attended) focused on food environments and policy rather than individual behaviours. A fair few presenters and discussants had lived experience of obesity. They used the terms, “overweight” and “obese”. It would be interesting to know whether they considered these terms stigmatising, and simply used them because everyone else at the conference was/in the literature does, or what alternative terms they would prefer. In any case, we have made these changes in the manuscript. 

Line 76: clarify that you mean ‘anti-obesity’ rather than ‘obesity’

Authors’ response:

We have changed this to “reducing obesity” and “improving health and wellbeing”

Line 89: there was a representative sampling of New Zealand

Authors’ response:

We assume you are referring to Madden (2012)? We don’t believe there is any indication of rate of intuitive eating. 

Line 124: r’ates of people with a BMI>30kg/m2’ or ‘higher body weight is more prevalent’

Authors’ response:

We have made this change.

Line 129: MM’s own relationship with his body and whether he perceived that he used eating behaviours to control his body size is relevant. Did his body mass exceed BMI25kg/m2 at the time of the interviews? Did he personally ascribe to intuitive eating? It is not possible to be bias-free in this context. His body size may also have influenced the participant’s responses.

Authors’ response:

We have added this information.

Line 154: was the coding guide and theme identification corroborated by another researcher? Why, why not?

Authors’ response:

We have added the following sentences to the Data Analysis section: “The results were discussed with the other co-authors and the first author also read the transcripts. All three authors agreed with the findings.”

Line 174-191: this relates to focus group protocol and should be located within methodology rather than results.

Authors’ response:

Agreed. This section has been moved.

Line 184 – Intuitive Eating does not claim to be a weight management strategy or effective at reaching an ‘appropriate weight’ – which participants might have interpreted as lower BMI as per social norms. Limitations should state that this could have mislead participants.

Authors’ response:

This is a valid point and one we had not considered. This has been added to the limitations section. 

Line 543: ‘be’ should be ‘with’ or similar

Authors’ response:

Thanks for spotting this typo. We have changed this to “connections with better health”.

---

## [Decision Letter · Decision Letter 1]

21 Jun 2023

PONE-D-22-32682R1What do people think of intuitive eating?  A qualitative exploration with rural AustraliansPLOS ONE

Dear Dr. Van Dyke,

Thank you for submitting your manuscript to PLOS ONE. After careful consideration, we feel that it has merit but does not fully meet PLOS ONE’s publication criteria as it currently stands. Therefore, we invite you to submit a revised version of the manuscript that addresses the points raised during the review process.

 Please note that PLOS ONE does not provide copyediting or proofs of accepted manuscripts. We therefore recommend that you carefully review your manuscript and correct any language errors at this time.

We look forward to receiving your revised manuscript.

Kind regards,

Jianhong Zhou

Staff Editor

PLOS ONE

Journal Requirements:

Reviewers' comments:

Reviewer's Responses to Questions

**Comments to the Author**

1. If the authors have adequately addressed your comments raised in a previous round of review and you feel that this manuscript is now acceptable for publication, you may indicate that here to bypass the “Comments to the Author” section, enter your conflict of interest statement in the “Confidential to Editor” section, and submit your "Accept" recommendation.

Reviewer #1: All comments have been addressed

2. Is the manuscript technically sound, and do the data support the conclusions?

Reviewer #1: (No Response)

3. Has the statistical analysis been performed appropriately and rigorously? 

Reviewer #1: (No Response)

4. Have the authors made all data underlying the findings in their manuscript fully available?

Reviewer #1: (No Response)

5. Is the manuscript presented in an intelligible fashion and written in standard English?

Reviewer #1: (No Response)

6. Review Comments to the Author

Reviewer #1: I thank the authors for addressing all of my previous comments. I found a couple of typographical errors the authors may want to fix should the manuscript be accepted for publication (line numbers refer to track changes version):

Line 62: BMI should be written out in full first here, then on lines 78 and 83 just use the abbreviation rather than spelling out in full again.

Line 114 missing ) after (e.g., [13-15

Line 598 is ‘(41)’ a reference? It differs in style to the other in-text citations which use square brackets

7. PLOS authors have the option to publish the peer review history of their article (what does this mean?). If published, this will include your full peer review and any attached files.

Reviewer #1: No

---

## [Author Response · Author response to Decision Letter 1]

3 Jul 2023

Line 62: BMI should be written out in full first here, then on lines 78 and 83 just use the abbreviation rather than spelling out in full again.

Author response: This correction has been made.

Line 114 missing ) after (e.g., [13-15

Author response: This correction has been made.

Line 598 is ‘(41)’ a reference? It differs in style to the other in-text citations which use square brackets"

Author response: This correction has been made. Please note that the correct reference number is 42. This correction has been made.

---

## [Editor Report · Decision Letter 2]

31 Jul 2023

What do people think of intuitive eating?  A qualitative exploration with rural Australians

PONE-D-22-32682R2

Dear Dr. Van Dyke,

We’re pleased to inform you that your manuscript has been judged scientifically suitable for publication and will be formally accepted for publication once it meets all outstanding technical requirements.

Kind regards,

Jianhong Zhou

Staff Editor

PLOS ONE
---

## [Editor Report · Acceptance letter]

9 Aug 2023

PONE-D-22-32682R2 

What do people think of intuitive eating?  A qualitative exploration with rural Australians 

Dear Dr. Van Dyke:

I'm pleased to inform you that your manuscript has been deemed suitable for publication in PLOS ONE. Congratulations! Your manuscript is now with our production department. 

Kind regards, 

on behalf of

Jianhong Zhou 

Staff Editor

PLOS ONE